# Preference Based Adaptation for Learning Objectives

**Yao-Xiang Ding**    **Zhi-Hua Zhou**
National Key Laboratory for Novel Software Technology,
Nanjing University, Nanjing, 210023, China
{dingyx, zhouzh}@lamda.nju.edu.cn

## Abstract

In many real-world learning tasks, it is hard to directly optimize the true performance measures, meanwhile choosing the right surrogate objectives is also difficult. Under this situation, it is desirable to incorporate an *optimization of objective* process into the learning loop based on weak modeling of the relationship between the true measure and the objective. In this work, we discuss the task of objective adaptation, in which the learner iteratively adapts the learning objective to the underlying true objective based on the preference feedback from an oracle. We show that when the objective can be linearly parameterized, this preference based learning problem can be solved by utilizing the dueling bandit model. A novel sampling based algorithm DL$^2$M is proposed to learn the optimal parameter, which enjoys strong theoretical guarantees and efficient empirical performance. To avoid learning a hypothesis from scratch after each objective function update, a boosting based hypothesis adaptation approach is proposed to efficiently adapt any pre-learned element hypotheses to the current objective. We apply the overall approach to multi-label learning, and show that the proposed approach achieves significant performance under various multi-label performance measures.

## 1   Introduction

Machine learning approaches have already been applied on many real-world tasks, in which the target is usually to optimize some task-specific performance measures. For complex problems, the performance measures are usually hard to be optimized directly, such as the click-through-rate in online advertisement and the profit gain in recommendation system design. Instead of directly optimizing these complex measures, surrogate objectives with better mathematical properties are designed to simplify optimization. It is obvious that whether the objective is correctly designed essentially affects the application performance. However, it also requires delicate knowledge on the relationship between the true measure and the objective, which is sometimes difficult and challenging to acquire. Under this situation, it is more desirable to learn both objective and hypothesis simultaneously.

Based on this motivation, we consider the novel scenario of learning with objective adaptation from preference feedback. Under this scenario, in each iteration of the objective adaptation process, the learner maintains a pair of objective functions, as well as the corresponding learned hypotheses, obtained from the latest two iterations. An oracle then provides a preference over the pair of hypotheses to the learner, according to the true task performance measure. Based on this preference information, the learner updates both the objective function and the corresponding hypothesis. In special, this formulation even allows us to model complex scenerios when the true performance measure is not quantified, such as subjective human preference. It is expected that the objective function converges to the optimal one so that the learned hypothesis optimizes the true performance measure. In this work, we focus on the following linear parameterized objective function class. Denote the objective by $L^{\mathbf{w}}, \mathbf{w} \in \mathcal{W}$, in which $\mathcal{W}$ is the parameter space, and $\mathbf{w} = [w^1 \ w^2 \ \cdots \ w^K]$ is a $K$

dimensional real-valued vector. We assume that $L^{\mathbf{w}}$ can be represented as

$$L^{\mathbf{w}} = \sum_{i=1}^{K} w^i l^i + w^0 \Lambda, \quad w^0 \in \{0, 1\}, \tag{1}$$

in which $l^1, l^2, \ldots, l^K$ are $K$ convex element objectives, and $w^0$ is an additional indicator of $\Lambda$. When $w^0 = 0$, $L^{\mathbf{w}}$ is a linear combination of the $K$ element objectives. When $w^0 = 1$, $\Lambda$ can be utilized to represent an additional convex regularization term. It is easy to see that this linear formulation covers a broad class of commonly used objectives in different learning tasks. By choosing different $\mathbf{w}$, we are allowed to consider different trade-offs among element objectives. The target of objective adaptation is then to learn the optimal $\mathbf{w}_*$ which corresponds to the best trade-off leading to the optimal task performance measure. To ensure the problem of learning a hypothesis under any choice of $\mathbf{w}$ is solvable, we restrict $\forall w^i \geq 0$. When $w^0 = 0$, the scale of $\mathbf{w}$ does not matter, thus we restrict $\mathcal{W}$ to be the non-negative part of the $K$-dimensional unit sphere, i.e. $\|\mathbf{w}\|_2 = 1, \forall w^i \geq 0$. When $w^0 = 1$, i.e. the scale of $\mathbf{w}$ is meaningful, we restrict $\mathcal{W}$ to be the $K$-dimensional ball $\|\mathbf{w} - R\mathbf{1}^K\|_2 \leq R$, in which $\mathbf{1}^K$ is the $K$-dimensional full-one vector and $R$ is the radius. By this way, $L^{\mathbf{w}}$ is kept convex over all $\mathbf{w} \in \mathcal{W}$.

There are two main challenges under the above objective adaptation scenario. One is to learn the objective function based on preference feedback from the oracle, which requires proper modeling of the preference feedback. Another is how hypothesis learning can be done efficiently without learning from scratch when the objective is updated. In this work, we take the first step towards the above two challenges. First, we naturally formulate the objective adaptation process into the dueling bandit model [Yue *et al.*, 2012], in which $\mathbf{w}$ is treated as the bandit arm and the oracle preference is treated as the reward. A novel sampling based algorithm $DL^2M$ , which stands for Dueling bandit Learning for Logit Model, is proposed for learning the optimal weight $\mathbf{w}_*$, which enjoys $\tilde{O}(K^{3/2}\sqrt{T})$ regret bound and efficient empirical performance. Second, by assuming to learn $K$ element hypotheses $f^i$ beforehand, which correspond to one-hot weights $\mathbf{w}^i, i \in [K]$ with only one non-zero $w^i$, a novel gradient boosting based approach named Adapt-Boost is proposed for adapting the element hypotheses to the hypothesis $h^{\mathbf{w}}$ corresponding to any $\mathbf{w}$. We apply the proposed objective adaptation approach to multi-label learning, and the experimental results show that our approach achieves significant performance for various multi-label performance measures.

## 2 Related Work

Some similarities exist between the objective adaptation scenerio and multi-objective optimization (MOO) [Deb, 2014]. Under both scenerios, multiple element objectives are considered, and the trade-offs among them should be properly dealt with. While in MOO, the target is to figure out the Pareto solutions reflecting different trade-offs instead of a single optimal solution defined by the oracle's preference. In fact, for our objective adaptation problem, it is also possible to utilize evolutionary algorithms instead of the proposed $DL^2M$ algorithm. While evolutionary algorithms are usually heuristic and theoretical guarantees are lacking. In [Agarwal *et al.*, 2014], the multi-objective decision making problem is considered. The target of the learner is to optimize all objectives by observing the actions provided by a mentor. There is a significant difference between their setting and ours since we focus on general learning tasks instead of decision making.

The proposed $DL^2M$ algorithm belongs to the family of continuous dueling bandit algorithms. In [Yue and Joachims, 2009], an online bandit gradient descent algorithm [Flaxman *et al.*, 2005] was proposed, which achieves $O(\sqrt{K}T^{3/4})$ regret bound for convex value functions. In [Kumagai, 2017], they showed that when the value function is strongly convex and smooth, their stochastic mirror descent algorithm achieves near optimal $\tilde{O}(K\sqrt{T})$ regret bound. Similar to $DL^2M$ , both the above two algorithms follow from the online convex optimization framework [Zinkevich, 2003], while $DL^2M$ assumes the underlying value function follows from a linear model. The major advantage of $DL^2M$ lies on the reduction of the total number of arms needed to be sampled during learning. For the above two algorithms, two arms are needed to be sampled for comparison in one iteration. While $DL^2M$ samples only one arm in one iteration $t$, and compares it with the arm sampled on $t-1$. Thus the total number of arms needed is halved for $DL^2M$ , comparing to the above two algorithms. For objective adaptation, choosing an arm incurs the cost of learning the corresponding hypothesis, thus it is important to reduce the total number of arms sampled.

Our boosting based hypothesis adaptation procedure is motivated from multi-task learning [Evgeniou and Pontil, 2004; Chapelle *et al.*, 2010]. By regarding each element objective as a single task, the hypothesis adaptation procedure can be decomposed into the adaptation of the *element hypothesis* for each element objective, and the adaptation of a *global hypothesis* for the weighted total objective. On the other hand, since the target is to optimize the single total objective, directly utilizing multi-task learning approaches is invalid to our problem. The hypothesis adaptation task is also considered in [Li *et al.*, 2013], in which an efficient adaptation approach is proposed under the assumption that the auxiliary hypothesis is a linear model. On the other hand, the linear model assumption also restricts the capacity of adaptation. To address this issue, our approach utilizes a gradient boosting based learner, which can use any weak hypothesis for adaptation, thus the optimization procedure can be more flexible and efficient.

## 3 Dueling Bandit Learning for Objective Adaptation

In this section, a dueling bandit algorithm $\mathrm{DL^2M}$ is proposed to learn the optimal weight vector $\mathbf{w}_*$ from preference feedback to solve the objective adaptation task. For convenience of optimization, we assume the arm space for $\mathrm{DL^2M}$ is the full $K$-dimensional unit sphere $\mathcal{W} : \|\mathbf{w}\|_2 = 1$. How to apply $\mathrm{DL^2M}$ on $\mathcal{W}$ defined in Section 1 is discussed in Remark 3 below. To model the preference of the oracle, we assume a total order $\preceq$ exists on $\mathcal{W}$. For $\mathbf{w}_* \in \mathcal{W}$, we have $\mathbf{w} \preceq \mathbf{w}_*, \forall \mathbf{w} \in \mathcal{W}$. Whenever the oracle is given an ordered pair $(\mathbf{w}, \mathbf{w}')$, the oracle gives the feedback of $r = 1$ if $\mathbf{w}' \preceq \mathbf{w}$, and $r = -1$ otherwise. To precisely model the partial order and how the oracle provides the preference information, we assume that each arm can be evaluated by a value function $v(\mathbf{w})$, such that $v(\mathbf{w}') \leq v(\mathbf{w}) \Leftrightarrow \mathbf{w}' \preceq \mathbf{w}$, and the preference feedback is generated by the probabilistic model considering the gap between $v(\mathbf{w})$ and $v(\mathbf{w}')$: $\Pr(r = 1) = \mu(v(\mathbf{w}) - v(\mathbf{w}'))$, in which $\mu(x)$ is a strictly increasing link function. In this paper, the logistic probability model $\mu(x) = 1/(1 + \exp(-x))$ is utilized, which is the common choice in related researches. The generation of preferences is also assumed to be independent of other parts of learning. In each iteration $t$ out of the total $T$ iterations, a pair $(\mathbf{w}_t, \mathbf{w}'_t)$ is submitted to the oracle for feedback. The target is to minimize the total (pseudo) regret

$$\Delta_T = \sum_{t=1}^{T} \mu(v(\mathbf{w}_*) - v(\mathbf{w}_t)) + \mu(v(\mathbf{w}_*) - v(\mathbf{w}'_t)).$$

If further restricts $\mathbf{w}'_t$ to be $\mathbf{w}_{t-1}$, then we can only consider the summation over $\mathbf{w}_t$. Furthermore, by observing that $\mu(v(\mathbf{w}_*) - v(\mathbf{w}_t))$ achieves minimum $1/2$ when $\mathbf{w}_t = \mathbf{w}_*$, we can reformulate the regret as $\Delta_T = \sum_{t=1}^{T} \mu(v(\mathbf{w}_*) - v(\mathbf{w}_t)) - \mu(v(\mathbf{w}_*) - v(\mathbf{w}_*)) = \sum_{t=1}^{T} \mu(v(\mathbf{w}_*) - v(\mathbf{w}_t)) - 1/2$. From the above definition, the tasks of regret minimization and optimal weight vector estimation coincide. By L'Hopital's rule, $f(x) = 1/(1+e^{-x}) - 1/2$ has the same convergence rate as $f(x) = x$ when $x \to 0$. Thus we can only consider $\Delta_T = \sum_{t=1}^{T} v(\mathbf{w}_*) - v(\mathbf{w}_t)$. In this work, we adopt the commonly used linear value function $v_{\mathrm{LIN}}(\mathbf{w}) = \mathbf{w}^T \theta_*$, in which $\theta_*$ is an underlying optimal evaluation vector. This leads to the classical linear regret formulation

$$\Delta_T^{LIN} = \sum_{t=1}^{T} \mathbf{w}_*^T \theta_* - \mathbf{w}_t^T \theta_*, \tag{2}$$

which indicates that the objective is to maximize the linear value function. Since $\|\mathbf{w}\|_2 = 1$, $\mathbf{w}^T \theta_*$ is the projection of $\theta_*$ onto $\mathbf{w}$, and achieves the maximum when the directions of $\mathbf{w}$ and $\theta_*$ coincide. Thus the direction of $\theta_*$ can be interpreted as the direction of the optimal weight vector kept in the oracle's mind. To simplify optimization, we assume $\|\theta_*\|_2 \leq 1$ without loss of generality.

From the definition of regret, it is crucial to estimate $\theta_*$ accurately. Thus we consider the procedure of estimating $\theta_*$ in each iteration first. Motivated by the logit one-bit bandit algorithm proposed in [Zhang *et al.*, 2016], in each iteration $t$, we can utilize the online version of the maximum likelihood estimator, i.e. to minimize the loss function

$$f_t(\theta) = \log \left( 1 + \exp \left( - r_t(\mathbf{w}_t^T \theta - \mathbf{w}_{t-1}^T \theta) \right) \right),$$

---

**Algorithm 1** Dueling bandit Learning for Logit Model (DL$^2$M )

---
1: **Input** Initialization $\theta_1 = 0, Z_1 = \lambda I, \mathbf{w}_0$, number of iterations $T$.
2: **for** $t = 1$ **to** $T$ **do**
3:    Sample $\eta_t \sim \mathcal{N}(0, I^K)$.
4:    Choose $\kappa_t$ according to Theorem 1.
5:    Compute $\tilde{\theta}_t$ as

$$\tilde{\theta}_t \leftarrow \theta_t + \kappa_t Z_t^{-1/2} \eta_t. \tag{5}$$

6:    Compute $\mathbf{w}_t$ as

$$\mathbf{w}_t \leftarrow \arg \max_{\|\mathbf{w}\|_2 = 1} \mathbf{w}^T \tilde{\theta}_t. \tag{6}$$

7:    Submit $\mathbf{w}_t$ and $\mathbf{w}_{t-1}$ and get $r_t$.
8:    Compute $\theta_{t+1}$ and $Z_{t+1}$ as Equation 3 and 4.
9: **end for**

---

which satisfies the exponentially concave property. As a result, the optimal update can be approximated by the analogy of the online Newton step [Hazan *et al.*, 2007]:

$$\theta_{t+1} = \min_{\|\theta\|_2 \leq 1} \frac{\|\theta - \theta_t\|_{Z_{t+1}}^2}{2} + (\theta - \theta_t)^T \nabla f_t(\theta_t), \tag{3}$$

in which

$$Z_{t+1} = Z_t + \frac{\beta}{2}(\mathbf{w}_t - \mathbf{w}_{t-1})(\mathbf{w}_t - \mathbf{w}_{t-1})^T, \quad Z_1 = \lambda I, \tag{4}$$

and $\beta = \frac{1}{2(e+1)}$. Next, we consider how to choose $\mathbf{w}_t$ in each round for better exploration. Different from the UCB based exploration strategy implemented in [Zhang *et al.*, 2016], we extend the linear Thompson sampling technique proposed in [Abeille and Lazaric, 2017] to our dueling bandit setting, leading to the DL$^2$M algorithm, which is illustrated in Algorithm 1. We provide regret guarantee for the proposed algorithm, whose proof will be presented in a longer version of the paper.

**Theorem 1.** *Assume that $\kappa_t$ in Algorithm 1 is set according to $\kappa_t = \sqrt{\gamma_t(\frac{\delta}{4T})}$, where*

$$\gamma_{t+1}(\delta) = \lambda + 16 + (\frac{8}{\beta} + \frac{32}{3}) \log \left( \frac{2\lceil 2\log t \rceil t^2}{\delta} \right) + \frac{2}{\beta} \log \frac{\det(Z_{t+1})}{\det(Z_1)}. \tag{7}$$

*After running DL$^2$M for $T$ rounds, then for $\forall \delta > 0$, the following result holds with probability at least $1 - \delta$:*

$$\sum_{t=1}^{T} \mathbf{w}_*^T \theta_* - \mathbf{w}_t^T \theta_* \leq \sqrt{\gamma_T(\frac{\delta}{4T}) KT \log \frac{8KT}{\delta}} \left( 391 \sqrt{\frac{1}{\beta} \log \frac{\det(Z_{t+1})}{\det(Z_1)}} + 128 \sqrt{\frac{1}{\lambda} \log \frac{4}{\delta}} \right).$$

By Lemma 10 of [Abbasi-Yadkori *et al.*, 2011], we have $\log(\det(Z_{t+1})/\det(Z_1)) \leq K \log \left( 1 + \frac{\beta t}{2\lambda K} \right)$. Thus Theorem 1 provides $\tilde{O}(K^{3/2}\sqrt{T})$ regret guarantee for DL$^2$M .

**Remark 1** The well-known doubling trick [Shalev-Shwartz, 2012] can be utilized to make $\kappa_t$ independent of the total number of iterations $T$. In practice, since $\kappa_t$ determines the step size of exploration, it is desirable to further make it fine-tunable. In all the experiments, we set $\kappa_t = \min(c/2, c\sqrt{\log(\det(Z_t)/\det(Z_1))})$, in which $c$ is a hyperparameter. The $\min$ operator is introduced to control the largest step size.

**Remark 2** Theorem 1 provides the guarantee of total regret. Our objective adaptation approach can be utilized in many real-world tasks, in which the learned is already in application during the learning stage, and the preference feedback is generated from its true effectiveness. The total regret guarantee is natural under this situation. Meanwhile, it is also important to consider another kind of tasks, in which only the final estimation accuracy matters. Under this situation, it is better to consider simple regret instead of total regret since it is a pure exploration problem. This is a particularly interesting

and challenging task since we assume the continuous arm space. The experimental results in Section 5 show that DL$^2$M is efficient in finding the best arms, and we leave designing the optimal pure exploration algorithm for continuous dueling bandits a future work to investigate.

**Remark 3** In the above discussion, we assume that the arm space for DL$^2$M is $\mathcal{W}_0 : \|\mathbf{w}\|_2 = 1$. For objective learning, the parameter domains introduced in Section 1 are different. We discuss how DL$^2$M can be applied.

When $w^0 = 0$, the domain of $\mathbf{w}$ is $\mathcal{W} : \|\mathbf{w}\|_2 = 1, \forall w^i \geq 0$, which is the nonnegative part of $\mathcal{W}_0$. To apply DL$^2$M , it is necessary to restrict each dimension of $\theta_t, \tilde{\theta}_t$ to be nonnegative. For $\theta_t$, we can simply change the domain of the update of $\theta$ as

$$\theta_{t+1} = \min_{\|\theta\|_2 \leq 1, \forall \theta^i \geq 0, i \in [K]} \frac{\|\theta - \theta_t\|^2_{Z_{t+1}}}{2} + (\theta - \theta_t)^T \nabla f_t(\theta_t),$$

in which $\theta^i$ is the $i$-th entry of $\theta$. Since the domain remains convex, the efficiency of optimization is unaffected. For $\tilde{\theta}_{t+1}$, we can simply take a $\tilde{\theta}^i_{t+1} \leftarrow \max(0, \tilde{\theta}^i_{t+1}), \forall i \in [K]$ operation to limit its value. Though this operation may affect the theoretical guarantee for $\mathbf{w}$ near the boundary, we observe that the performance is not affected in experiments.

When $w^0 = 1$, the domain of $\mathbf{w}$ is $\mathcal{W} : \|\mathbf{w} - R\mathbf{1}^K\|_2 \leq R$. The main idea is to establish a topologically identical mapping from the arm space to $\mathcal{W}$, then we can perform DL$^2$M in the arm space, then map the result to $\mathcal{W}$. First, it is easy to establish a bijective mapping $g_1$ from the $K$ dimensional ball $\mathcal{W}_1 : \|\mathbf{w}\|_2 \leq 1$ to $\mathcal{W}$ with constant shifting and scaling. Second, another simple bijective mapping $g_2$ exists to map a point in half of the $K + 1$-dimensional sphere, i.e. $\mathcal{W}_2 : \|\mathbf{w}\|_2 = 1, w^{K+1} \geq 0$, to a point in $\mathcal{W}_1$, by simply setting $w^{K+1} = 0$ (just imagine the mapping from the upper half of the three-dimensionl sphere onto a two-dimensional circle). Thus we can simply utilize the composite mapping $g_1(g_2)$ to map an arm in $\mathcal{W}_2$ to a parameter in $\mathcal{W}$. To apply DL$^2$M on $\mathcal{W}_2$, we can update $\theta$ by

$$\theta_{t+1} = \min_{\|\theta\|_2 \leq 1, \theta^{K+1} \geq 0} \frac{\|\theta - \theta_t\|^2_{Z_{t+1}}}{2} + (\theta - \theta_t)^T \nabla f_t(\theta_t),$$

and perform $\tilde{\theta}^{K+1}_{t+1} \leftarrow \max(0, \tilde{\theta}^{K+1}_{t+1})$ to restrict both $\theta_{t+1}, \tilde{\theta}_{t+1}$, which is similar to $w^0 = 0$.

## 4    Boosting Based Hypothesis Adaptation

After each objective adaptation step, $\mathbf{w}$ is updated, then a new $L^{\mathbf{w}}$ is obtained. To avoid learning the corresponding hypothesis $F^{\mathbf{w}}$ from scratch, a hypothesis adaptation procedure is considered. Recall the formulation of objective function defined in Equation 1. Assume that before the whole objective adaptation process, we have learned $K$ element hypotheses $f^i$ under (regularized) element objectives $l^i + w^0 \Lambda, i \in [K]$. To obtain $F^{\mathbf{w}}$ corresponding to $L^{\mathbf{w}}$, we can linearly combine $f^i$ together with a newly learned auxiliary hypothesis $\phi^{\mathbf{w}}$, i.e. make $F^{\mathbf{w}} = \sum_{i=1}^K \alpha^i f^i + \phi^{\mathbf{w}}$. As a result, the learning problem is transformed into

$$\min_{\alpha^i, i \in [K], \phi^{\mathbf{w}}} L^{\mathbf{w}} \left( \left( \sum_{i=1}^K \alpha^i f^i \right) + \phi^{\mathbf{w}} \right). \tag{8}$$

Under the above formulation, the learning target is to decide the weight $\alpha^i$ for each $f^i$, together with the auxiliary hypothesis $\phi^{\mathbf{w}}$. Intuitively, there should be a close relationship between $\alpha^i$ and $w^i$, which is the weight for $l^i$ in $L^{\mathbf{w}}$. When $w^i$ is large, the corresponding $l^i$ has a large impact to the global $L^{\mathbf{w}}$. Since $f^i$ is learned under $l^i + w^0 \Lambda$, then $\alpha^i$ should also be large to make the contribution of $f^i$ in $F^{\mathbf{w}}$ more significant. For the similar reason, if $w^i$ is small then $\alpha^i$ should follow. As a result, to solve Equation 8 properly, establishing a close relationship between $\alpha^i$ and $w^i$ is a necessary task.

Based on this motivation, a boosting based hypothesis adaptation approach named Adapt-Boost is proposed. Assume that the learning procedure runs for $N$ iterations. Under Adapt-Boost, one weak hypothesis $h^j$ is learned in each iteration $j$. Denote by $H = [h_1 \ h_2 \ \cdots \ h_N]^T$ the vector of all learned weak hypotheses and set $w'^{,0} = 1, w'^{,i} = w^i, i \in [K]$, Adapt-Boost solves the following

---

**Algorithm 2** Adapt-Boost

---

1: **Input**: Loss parameter $\mathbf{w}$, loss function $L^{\mathbf{w}}$, element hypotheses $f^i, i \in [K]$, $f^0 \equiv 0$, number of iterations $N$, number of element losses $K$, step size $\epsilon$.
2: $w'^{,i} \leftarrow w^i, i \in [K], w'^{,0} \leftarrow 1, F_0 \leftarrow f^0$.
3: **for** $j = 1$ **to** $N$ **do**
4:     Calculate current residual $-\nabla L^{\mathbf{w}}(F_{j-1})$.
5:     **for** $i = 0$ **to** $K$ **do**
6:         Fit residual $-\nabla L^{\mathbf{w}}(F_{j-1})$ with $f^i + h^i_j$ to obtain a weak hypothesis $h^i_j$.
7:     **end for**
8:     Choose the optimal update $i^*$ as

$$i^* = \arg\max_i -w'^{,i}[\nabla L^{\mathbf{w}}(F_{j-1})](f^i + h^i_j).$$

9:     Update the current hypothesis as

$$F_j = F_{j-1} + (w'^{,i^*}\epsilon)(f^{i^*} + h^{i^*}_j).$$

10: **end for**
11: **Output** The learned hypothesis $F_N$.

---

$l_1$-regularized problem:

$$\min_{\beta^i, i \in [K] \cup \{0\}, H} L^{\mathbf{w}}\left(\left(\sum_{k=1}^{K} \beta^{k,T}(\mathbf{1}^N f^k + H)\right) + \beta^{0,T}H\right), \quad s.t. \sum_{i=0}^{K} \frac{1}{w'^{,i}}\|\beta^i\|_1 \leq \mu, \quad (9)$$

in which $\beta^i, i \in [K] \cup \{0\}$ are $N$-dimensional weight vectors and $\mathbf{1}^N$ is the $N$-dimensional full-one vector. Comparing to Equation 8, $F^{\mathbf{w}}$ is further restricted as $\left(\sum_{k=1}^{K} \beta^{k,T}(\mathbf{1}^N f^k + H)\right) + \beta^{0,T}H$, and the auxiliary hypothesis $\phi^{\mathbf{w}}$ is decomposed into $K$ *local* $\beta^{k,T}H$ corresponding to $f^k$, together with a *global* $\beta^{0,T}H$. For each $f^k$, the weight $\alpha^k$, which represents the importance of $f^k$ in learning $F^{\mathbf{w}}$, is substituted by the weight vector $\beta^k$. Thus controlling the magnitude of $\alpha^k$ is equivalent to controlling the norm of $\beta^k$. This target is realized by introducing the sum of $1/w'^{,i}$-weighted $l_1$-norm constraints on $\beta^i, i \in [K] \cup \{0\}$ in Equation 9 with a hyperparameter $\mu$ controlling the global sparsity. Meanwhile, by controlling the local sparsity of each $\beta^k$ using $w'^{,k}, k \in [K]$, we are able to relate the importance of $f^k$ in $F^{\mathbf{w}}$ with objective weights $\mathbf{w}$.

The key advantage to employ Equation 9 is that this sparsity-constrained problem can be solved by the $\epsilon$-boost algorithm [Rosset *et al.*, 2004], which will be briefly introduced below. To simplify notation, we use $\beta$ to denote the vector which is the concatenation of all $\beta^i, i \in [K] \cup \{0\}$. Temporarily, we also assume that the weak hypotheses $H$ are fixed, and only $\beta$ needs to be optimized. Instead of explicitly setting the sparsity level $\mu$, we decompose the sparsity constraint over all steps. In each iteration, a small increment $\Delta\beta$ is added on $\beta$, and an $\epsilon$-sparsity constraint is applied on $\Delta\beta$, leading to the following inside-iteration optimization problem:

$$\min_{\Delta\beta} L^{\mathbf{w}}(\beta + \Delta\beta), \quad s.t. \sum_{i=0}^{K} \frac{1}{w'^{,i}}\|\Delta\beta^i\|_1 \leq \epsilon, \quad (10)$$

in which $\Delta\beta^i$ is the part of $\Delta\beta$ added on $\beta^i$. The objective function can be approximated as

$$L^{\mathbf{w}}(\beta + \Delta\beta) \approx L^{\mathbf{w}}(\beta) + [\nabla L^{\mathbf{w}}(\beta)]^T \Delta\beta \quad (11)$$

by Taylor expansion. Thus we turn to minimize $[\nabla L^{\mathbf{w}}(\beta)]^T \Delta\beta$. Since the sparsity constraints are gradually added by $\epsilon$ over the learning process, and $L^{\mathbf{w}}$ is convex, the optimal solution for Equation 10 always satisfies $\sum_{i=0}^{K} \frac{1}{w'^{,i}}\|\Delta\beta^i\|_1 = \epsilon$. Let $[\nabla L^{\mathbf{w}}(\beta)]^i_j, [\Delta\beta]^i_j, i \in [K] \cup \{0\}, j \in [N]$ be the $(Ni + j)$-th dimension of $\nabla L^{\mathbf{w}}(\beta)$ and $\Delta\beta$. It is easy to see that the optimal $\Delta\beta$ in Equation 11 is a vector of all zeros except for $[\Delta\beta]^{i^*}_{j^*} = w'^{,i^*}\epsilon$ such that $i^*, j^* = \arg\min_{i,j} w'^{,i}[\nabla L^{\mathbf{w}}(\beta)]^i_j$. Furthermore, we can explicity write the $(Ni + j)$-th component of $\nabla L^{\mathbf{w}}(\beta)$ as $\partial L^{\mathbf{w}}/\partial \beta^i_j = [\nabla L^{\mathbf{w}}(F^{\mathbf{w}})](\partial F^{\mathbf{w}}/\partial \beta^i_j) = [\nabla L^{\mathbf{w}}(F^{\mathbf{w}})](f^i + h_j)$, in which $h_j$ is the $j$-th weak hypothesis in

$H$, and an additional $f^0 \equiv 0$ is introduced to simplify notation. Now let us take choosing weak hypotheses $H$ into consideration. Based on the above discussion, in the $j$-th iteration, our target is to solve $\max_{i,h_j} -w'^{,i}[\nabla L^{\mathbf{w}}(F^{\mathbf{w}})](f^i + h_j)$. This formulation inspires us to utilize gradient boosting. To obtain the optimal update, a candidate weak hypothesis $h_j^i$ is chosen for each $i, i \in [K] \cup \{0\}$ to let $f^i + h_j^i$ fit for the residual $-\nabla L^{\mathbf{w}}(F^{\mathbf{w}})$, and then we can choose the optimal update $f^{i^*} + h_j^{i^*}$ which optimally fits the residual weighted by $w'^{,i}$. The optimal step size for the update is $w'^{,i^*} \epsilon$ according to the previous discussion. The whole process of Adapt-Boost is illustrated in Algorithm 2. It can be seen that Adapt-Boost utilizes a boosting based process to gradually add the element and weak hypotheses into the learned hypothesis instead of explicitly setting their weights. Thanks to the flexability of choosing the weak learners and the efficiency of gradient boosting, we are able to solve complex hypothesis adaptation problems with low cost.

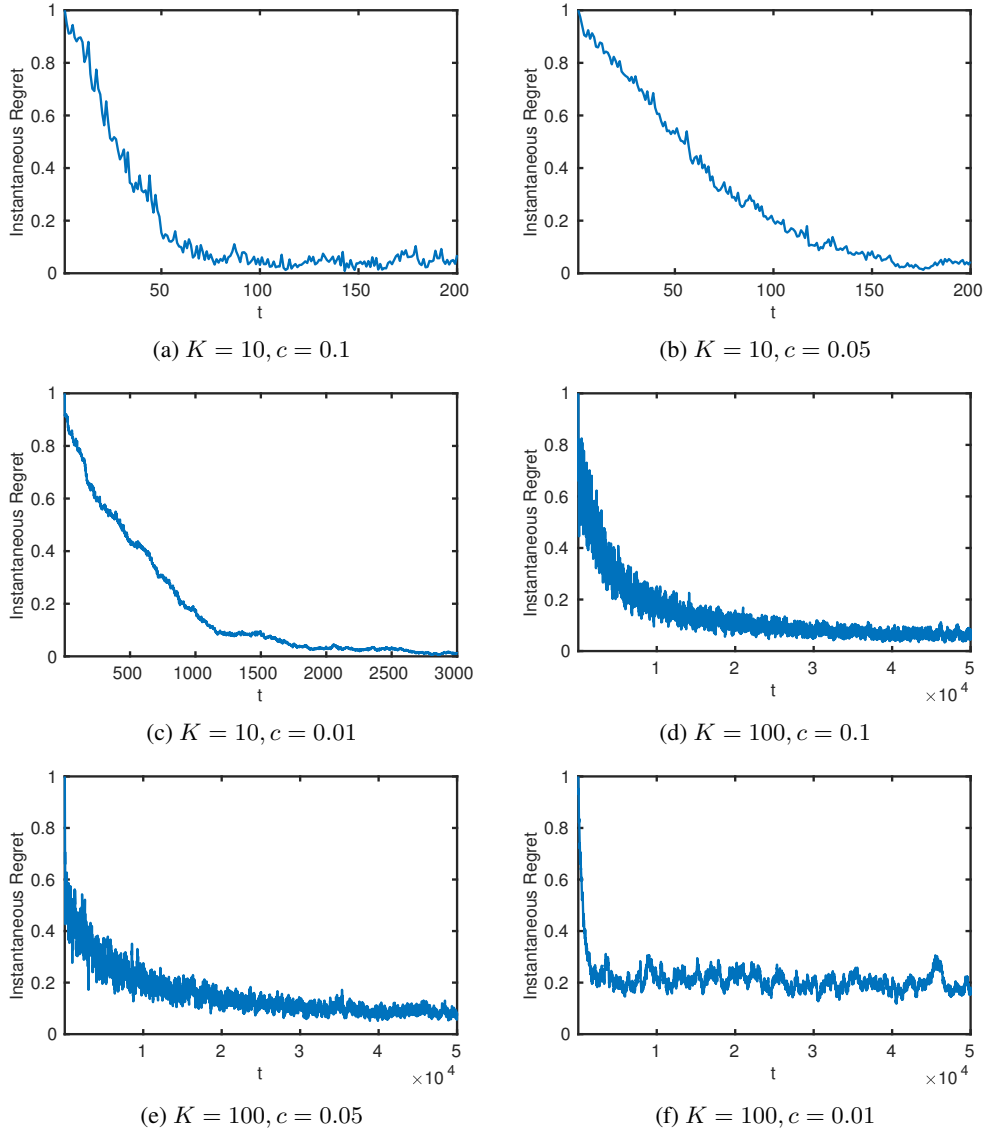

(a) $K = 10, c = 0.1$

(b) $K = 10, c = 0.05$

(c) $K = 10, c = 0.01$

(d) $K = 100, c = 0.1$

(e) $K = 100, c = 0.05$

(f) $K = 100, c = 0.01$

Figure 1: Instantaneous regret of DL$^2$M .

# 5 Experiments

## 5.1 Testing DL$^2$M on Synthetic Data

We present experimental results on synthetic data to verify the effectiveness of DL$^2$M . In each experiment, a $K$ dimensional point is uniformly sampled from the unit ball as $\theta_*$. Once the learner submits the pair of arms $(\mathbf{w}_t, \mathbf{w}_{t-1})$, a preference feedback $r_t \in \{-1, 1\}$ is randomly generated according to $\Pr(r_t = \pm 1|(\mathbf{w}_t, \mathbf{w}_{t-1})) = 1/(1 + \exp\left(-\rho r_t(\mathbf{w}_t - \mathbf{w}_{t-1})^T \theta_*\right))$, in which $\rho$ is the parameter controlling the randomness of the preferences. In all experiments, we use $\rho = 100$ to ensure the preferences are relatively consistent. We also set $\lambda = 1$ in all the experiments. The performance is measured by the change of instantaneous regret $\mathbf{w}_*^T \theta_* - \mathbf{w}_t^T \theta_*$ over time. We compare the results among different $c$ and $K$, which are illustrated in Figure 1. It can be observed that when the parameter $c$ is properly set and $K$ is not large, DL$^2$M achieves very efficient performance, which can quickly converge in limited number of iterations. As the dimension gets larger, the performance degenerates accordingly. To verify the efficiency of Thompson sampling in dueling and one-bit bandit problems, it is interesting to compare the above results with those reported in [Zhang *et al.*, 2016], which utilizes UCB based exploration. It can be seen that Thompson sampling can achieve more efficient performance in practice as in many other bandit problems.

## 5.2 Multi-Label Performance Measure Adaptation

The multi-label classification task is utilized to evaluate the effectiveness of DL$^2$M and Adapt-Boost, both separately and jointly. It is well-known that for multi-label learning, various performance measures exist, and the choice of the performance measure will largely affect the evaluation of the learned classifier. In [Wu and Zhou, 2017], two notions of multi-label margin, i.e. label-wise and instance-wise margins are proposed to characterize different multi-label performance measures. According to their work, one specific multi-label performance measure tends to be biased towards one of the two margins, such that optimizing the corresponding margin will also make the measure optimized. Based on this finding, the stochastic gradient descent (SGD) based LIMO algorithm is proposed to jointly optimize the both margins, in order to achieve good performance on different measures simultaneously. The high-level formulation of LIMO's objective is

$$L_{LIMO} = \Lambda + w^1 L_{label} + w^2 L_{inst}, \tag{12}$$

in which $L_{label}, L_{inst}$ are two margin loss terms for maximizing the two margins, and $\Lambda$ is a regularization term. For $L_{LIMO}$, the weights $w^1, w^2$ control the relative importance of the two loss terms, thus different choices of the weights can significantly affect the performance. We will show that by utilizing DL$^2$M and Adapt-Boost, we can automatically find the proper weights between the two margin losses in LIMO's objective, and efficiently adapt to different performance measures.

The experiments are conducted on six benchmark multi-label datasets [1]: emotions, CAL500, enron, Corel5k, medical and bibtex. On each dataset, four multi-label performance measures are adopted for evaluation, i.e. ranking loss, coverage, average precision and one error. Three LIMO based comparison methods are adopted as baselines: (i) **LIMO-label**, which optimizes $L_{LIMO}$ with $w^1 = 1, w^2 = 0$, (ii) **LIMO-inst**, which optimizes $L_{LIMO}$ with $w^1 = 0, w^2 = 1$, (iii) **LIMO**, which optimizes $L_{LIMO}$ with $w^1 = 1, w^2 = 1$ and corresponds to the recommended parameter in the original paper. To evaluate DL$^2$M and Adapt-Boost both separately and jointly, three adaptation based approaches are tested: (i) **ADAPT-hypo**, which optimizes $L_{LIMO}$ with $w^1 = 1, w^2 = 1$ using Adapt-Boost, (ii) **ADAPT-obj**, which utilizes DL$^2$M with SGD training, (iii) **ADAPT-both**, which utilizes both DL$^2$M and Adapt-Boost. To implement DL$^2$M , each dataset is randomly split into training, validation and testing set, with ratio of size 3:1:1. During the learning process, the preference feedback is generated by testing the learned hypothesis on the validation set, and DL$^2$M is utilized to update the objective for 20 iterations, with $c = 0.05, \lambda = 1$. For Adapt-Boost, to evaluate its efficiency, we only use half number of training iterations than standard LIMO training. Furthermore, to make Adapt-Boost compatible to LIMO training, the SGD updates are utilized as the weaker learners for adaptation.

The experimental results are illustrated in Table 1, and the average ranks in all experiments are illustrated in Table 2. It can be seen that DL$^2$M based method ADAPT-obj achieve better performance than LIMO, which assigns fixed weights to the two margin losses. This phenomenon reveals

that $DL^2M$ can automatically identify the best trade-off among different element objectives. Furthermore, though running with much fewer training iterations, Adapt-Boost based method ADAPT-hypo achieves even better performance than LIMO, which is based on standard SGD training. This verifies the efficiency of Adapt-Boost. ADAPT-both, which utilizes both two adaptation methods, achieves superior performance. It shows that by utilizing $DL^2M$ and Adapt-Boost, we can effectively solve the objective and hypothesis adaptation problem better and faster.

| Dataset | Algorithm | ranking loss ↓ | coverage ↓ | avg. precision ↑ | one-error ↓ |
|---|---|---|---|---|---|
| emotions | LIMO-inst | .420 ± .051(6) | 2.950 ± .134(6) | .603 ± .028(6) | .500 ± .047(6) |
| | LIMO-label | .349 ± .028(5) | 2.745 ± .174(5) | .619 ± .025(5) | .509 ± .064(5) |
| | LIMO | .299 ± .023(4) | 2.483 ± .070(4) | .648 ± .028(4) | .498 ± .057(4) |
| | ADAPT-hypo | .279 ± .026(3) | 2.331 ± .090(2) | .671 ± .032(3) | .481 ± .048(3) |
| | ADAPT-obj | .268 ± .033(2) | 2.377 ± .144(3) | .673 ± .033(2) | .478 ± .062(2) |
| | ADAPT-both | **.254 ± .020(1)** | **2.298 ± .200(1)** | **.678 ± .028(1)** | **.465 ± .058(1)** |
| CAL500 | LIMO-inst | .522 ± .026(6) | 162.950 ± 2.417(6) | .153 ± .010(6) | .971 ± .019(6) |
| | LIMO-label | .182 ± .005(2) | 131.439 ± 1.764(5) | .496 ± .006(5) | .099 ± .026(2) |
| | LIMO | .182 ± .004(2) | 131.020 ± 1.697(2) | **.498 ± .006(1)** | .131 ± .053(5) |
| | ADAPT-hypo | .182 ± .005(2) | 131.297 ± 1.899(4) | .497 ± .007(2) | .128 ± .036(4) |
| | ADAPT-obj | .182 ± .005(2) | 131.088 ± 1.849(3) | .497 ± .007(2) | **.098 ± .028(1)** |
| | ADAPT-both | **.181 ± .004(1)** | **131.008 ± 2.072(1)** | .497 ± .008(2) | .107 ± .024(3) |
| enron | LIMO-inst | .229 ± .010(6) | 25.166 ± .957(6) | .504 ± .017(6) | .350 ± .043(6) |
| | LIMO-label | .087 ± .009(3) | 12.362 ± .612(5) | .672 ± .014(4) | **.235 ± .024(1)** |
| | LIMO | .089 ± .009(5) | 12.199 ± .625(4) | .670 ± .014(5) | .246 ± .029(3) |
| | ADAPT-hypo | .087 ± .009(3) | 12.060 ± .648(2) | .680 ± .013(2) | .246 ± .022(3) |
| | ADAPT-obj | .086 ± .009(2) | **12.049 ± .624(1)** | .675 ± .012(3) | .251 ± .022(5) |
| | ADAPT-both | **.085 ± .008(1)** | 12.066 ± .577(3) | **.683 ± .016(1)** | .242 ± .027(2) |
| Corel5k | LIMO-inst | .302 ± .006(6) | 188.785 ± 3.122(6) | .101 ± .006(6) | .893 ± .007(6) |
| | LIMO-label | .121 ± .005(3) | 106.920 ± 2.457(5) | **.281 ± .005(1)** | **.718 ± .017(1)** |
| | LIMO | .130 ± .004(5) | 104.465 ± 2.149(4) | .222 ± .006(5) | .793 ± .012(5) |
| | ADAPT-hypo | .121 ± .005(3) | 101.668 ± 2.141(3) | .252 ± .006(3) | .762 ± .014(3) |
| | ADAPT-obj | .118 ± .005(2) | 100.478 ± 3.376(2) | .247 ± .010(4) | .772 ± .013(4) |
| | ADAPT-both | **.114 ± .005(1)** | **98.880 ± 2.989(1)** | .280 ± .007(2) | .719 ± .016(2) |
| medical | LIMO-inst | .019 ± .005(2) | 1.781 ± .337(5) | .857 ± .020(5) | .192 ± .034(5) |
| | LIMO-label | .028 ± .004(6) | 2.326 ± .489(6) | .830 ± .020(6) | .216 ± .037(6) |
| | LIMO | .020 ± .005(4) | 1.563 ± .249(3) | .869 ± .026(2) | **.163 ± .028(1)** |
| | ADAPT-hypo | .021 ± .004(5) | 1.621 ± .246(4) | .863 ± .023(4) | .181 ± .031(4) |
| | ADAPT-obj | .019 ± .004(2) | 1.499 ± .340(2) | **.874 ± .021(1)** | .171 ± .034(2) |
| | ADAPT-both | **.018 ± .004(1)** | **1.447 ± .288(1)** | .866 ± .025(3) | .176 ± .036(3) |
| bibtex | LIMO-inst | .120 ± .003(6) | 32.751 ± 1.144(6) | .488 ± .007(6) | .486 ± .018(6) |
| | LIMO-label | .072 ± .003(5) | 20.460 ± .515(5) | .526 ± .007(5) | .440 ± .018(5) |
| | LIMO | .060 ± .002(4) | 17.648 ± .596(4) | .567 ± .007(4) | .395 ± .014(4) |
| | ADAPT-hypo | **.056 ± .002(1)** | **16.708 ± .430(1)** | .579 ± .007(2) | .384 ± .013(2) |
| | ADAPT-obj | .057 ± .003(3) | 17.119 ± .832(2) | .575 ± .006(3) | .391 ± .013(3) |
| | ADAPT-both | **.056 ± .002(1)** | 17.128 ± .590(3) | **.581 ± .006(1)** | **.383 ± .013(1)** |

Table 1: Experimental results for the adaptation based and LIMO based methods. For each measure, "↓" indicates "the smaller the better" and "↑" indicates "the larger the better". The results are shown in mean±std(rank) format calculated from ten repeated experiments. The rank is calculated from the mean. The smaller the rank, the better the performance. The first-ranked results are bolded.

| Algorithm | LIMO-inst | LIMO-label | LIMO | ADAPT-hypo | ADAPT-obj | ADAPT-both |
|---|---|---|---|---|---|---|
| avg. rank | 5.71 | 4.21 | 3.67 | 2.83 | 2.42 | 1.58 |

Table 2: The average performance rank in all experiments.

# 6 Conclusion and Future Work

In this work, the preference based objective adaptation task is studied. The $DL^2M$ algorithm is proposed under this setting, which can efficiently solve the objective adaptation problem based on the dueling bandit model. For better hypothesis adaptation, the Adapt-Boost method is proposed in order to adapt the pre-learned element classifiers to the new objective with low cost.

To further investigate the objective adaptation problem, it is possible to relax the linear combinaiton formulation of the objective function adopted in this work. We are also interested in applying the proposed approaches in other real-world problems, especially the tasks in which human expert feedback can be utilized. Furthermore, it is also interesting to investigate Adapt-Boost on problems with larger scale, as well as to study its theoretical guarantees.

## Acknowledgement

This research is supported by National Key R&D Program of China (2018YFB1004300), NSFC (61751306) and Collaborative Innovation Center of Novel Software Technology and Industrialization. Yao-Xiang Ding is supported by the Outstanding PhD Candidate Program of Nanjing University. The Authors would like to thank the anonymous reviewers for constructive suggestions, as well as Lijun Zhang, Ming Pang, Xi-Zhu Wu and Yichi Xiao for helpful discussions.

## Footnotes

[1] http://mulan.sourceforge.net/datasets-mlc.html

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
