[Reviews · NeurIPS 2018]

Reviewer 1



The paper proposes to learn a linear model for combining multiple objectives via an oracle that provides feedback via preferences. It formulates the problem as a dueling bandit problem and applies a simple adaptation of dueling bandit gradient descent to solve the problem. When training the underlying hypothesis that will optimize the current linear combination is expensive, the paper also proposes a boosting style algorithm that basically uses as base learners models that have been trained on each objective separately. Experiments on a multilabel problem show that the proposed approach can find good combinations of two objectives (instance margin and label margin) that will optimize different multilabel metrics. The paper seems to be a straightforward combination of dueling bandits and the paper by Zhang et. al. 2016 and seems very incremental. Using boosting for blending the base models is a cute idea for avoiding to refit the model on the new objective as it evolves but I still don't think that this paper has enough new material to meet the NIPS bar.

Reviewer 2



Summary: The authors consider the problem of optimizing the linear combination of multiple objective functions, where these objective functions are typically surrogate loss functions for machine learning tasks. In the problem setting, the decision maker explore-while-exploit the linear combination in a duel bandit setting, where in each time step the decision maker tests the two hypotheses generated from two linear combinations, and then the decision maker would receive the feedback on whether the first hypothesis is better or the second is better. The main contributions of the paper is the proposal of online algorithms for the duel bandit problem, where the preference on two tested hypotheses is modeled by a binary logistic choice model. In order to avoid retraining the hypothesis for every different linear combination, the authors adapt the boosting algorithm, which focuses on optimizing the mixture of K different hypotheses, where each hypothesis stem from optimizing one surrogate function. Major Comment: I find the paper quite interesting in terms of problem model and the analysis, and I am more inclined towards acceptance than rejection. The problem considered by the authors is interesting, since choosing the right mixture of surrogate loss functions is important for machine learning tasks in general. While such a problem of properly balancing loss functions has been considered in the literature, the differentiating feature of the authors’ problem model is the duel bandit feedback setting. I guess the authors should motivate more on such a setting, for example saying that the duel bandit feedback could be generated from a human agent who makes a choice on the prediction outcomes under two given hypotheses. Nevertheless, overall I still find the problem interesting and worth studying. The part on solving the duel bandit model seems novel and important to me, though in fact I am not too familiar with the duel bandit literature. The part on boosting very much stems from existing results on boosting. While the boosting part serves as a nice complement to the duel bandit results, the former does not seem to be the main contribution of the paper. The numerical results suggest that the algorithmic framework developed by the authors indeed improves the performance when multiple surrogate loss functions are required to be balanced. Minor Comments: Line 124: The authors should avoid using the notation \Delta_T in Equation 2, since it contradicts with the definition of \Delta_T in Lines 110, 115. Line 169: It would be useful to spell out what CAPO stands for.

Reviewer 3



The paper discusses a problem of adapting an optimization objective to a complex test objective. A complex test objective can either be hard to define (revenue, attrition, etc.) or hard to optimize directly (e.g. ranking losses), and it may not be clear which optimization objective (continuous, possibly convex) should be used. The paper proposes an approach that jointly optimizes the parameter according to an optimization objective and adapts the optimization objective to the complex test objective. The main idea is to consider multiple optimization objectives and find the best linear combination of them. The solution has two main components. The first assumes black box access to a solver of the optimization task and finds the linear combination of the different objectives best suited to the test objective. The second solves each of the optimization objectives efficiently given the previous solutions. The first component assumes oracle access to the test loss, and build a solution via the multi-armed bandit technology. The second component adapts methods from multi-task learning to solve the particular problem at hand. Novelty / Impact: The problem proposed is very common, and I am surprised it did not receive more attention. The authors bring an educated solution to this problem, and this type of approach is likely to be proven useful in many real-life problems. The solution proposed seems somewhat raw. The main idea of having two components for (1) tuning the optimization objective (2) efficiently solving each optimization problem by utilizing previous solutions feels like the right thing to do. The details probably have room for improvement (see details below). That being said, the problem doesn’t have to be completely solved by a single paper, and I think the impact of the paper is still significant based on the importance of the problem and the clean high level solution Writing / Clarity: There are some parts of the paper that could be improved. There are many definitions that are not here, but can only be obtained by reading the cited papers. The motivation, at least for algorithm 2 can be better explained, especially given that the algorithm does not come with any guarantees. Furthermore, if one would like to implement algorithm 2 based on the paper, they would have a very hard time doing so. The authors should make a considerable effort to explain exactly how the algorithm works. Experiments: The experiments contain one experiment measuring the bandit algorithm (component 1) and another for the full system. In the section discussing the full system, the authors use the test set for measuring the test objective in the bandit algorithm. Thought claiming it does not leak information, it does, exactly as in the case of hyper parameter optimization. A cleaner experiment could have been done with a validation set used to tune the optimization objective. Another part that seems to be missing is an experiment measuring the effectiveness of the 2nd component without the first. For example, given the solutions to weights w_1,..,w_n, apply algorithm 2 to solve a new objective, and compare its performance to a solution obtained directly. Given that algorithm 2 doesn’t come with guarantees, that experiment would be more useful than experiments showing that algorithm 1 works. Overall the experiments paint an OK picture but can be made much more convincing. Summary: The paper tackles an important problem, its high level solution is clean and elegant, but the details leave room for improvement. The experiments are probably sufficient but not much more. Still, everything considered the fixes are doable within the time for the camera ready version and the advantages of the paper out-weigh its disadvantages. Comments: * For algorithm 1, the guarantee is for the regret, but the objective is clearly simple regret, i.e. identifying the best linear function. The authors should consider either presenting a different proof, or even adapting the algorithm with that in mind. In particular, the discussion about the UCB approach not working is not surprising as UCB may not work well for best arm identification. * In line 151 there is an explanation of how to adapt from the unit ball to positive numbers. A naive implementation of what is written may take time exponential in the dimension (drawing a random vector until all its components are positive). Please provide a more detailed explanation of how this can be done efficiently. Generally the explanation in the paragraph about the implementation is very vague and short. Perhaps add a detailed section in the appendix. * line 163. The parameter l^i seems to come out of nowhere * line 214: the parameter \kappa in alg. 1 is a matrix, but in the experiments it’s fixed as a constant number. Please fix this issue. Also, compare the method to a parameter kappa that is set according to what the algorithm dictates, and not just a constant. The choice of setting it as a constant is not clear, and is even shown to be wrong by the experiments. * Equation (13): Please define the the paper all the optimization objectives * Table 1: Please bold the best experiments in each setting